# Procrastination and Stress: A Conceptual Review of Why Context Matters

**DOI:** 10.3390/ijerph20065031

**Published:** 2023-03-13

**Authors:** Fuschia M. Sirois

**Affiliations:** Department of Psychology, Durham University, Durham DH1 3LE, UK; fuschia.sirois@durham.ac.uk; Tel: +44-(0)-114-222-6552

**Keywords:** procrastination, stress, COVID-19, personality, emotion regulation, coping

## Abstract

Research over the past two decades has continued to highlight the robust associations between procrastination and stress across multiple populations and contexts. Despite this burgeoning evidence base and theory linking procrastination to higher levels of stress, as well as the reverse, the role of context in this potentially dynamic association has received relatively little attention. In this conceptual review I argue that from a mood regulation perspective of procrastination, stressful contexts necessarily increase risk for procrastination because they deplete coping resources and lower the threshold for tolerating negative emotions. Drawing on insights from coping and emotion regulation theory, the new stress context vulnerability model of procrastination proposes that the risk for procrastination increases in stressful contexts primarily because procrastination is a low-resource means of avoiding aversive and difficult task-related emotions. The new model is then applied to evidence on the primary and secondary sources of stress during the COVID-19 pandemic and how they may have increased vulnerability for procrastination. After discussing potential applications of the new model for understanding how and why risk for procrastination may increase in other stressful contexts, approaches that might mitigate vulnerability for procrastination in high-stress contexts are discussed. Overall, this new stress context vulnerability model underscores the need for taking a more compassionate view of the antecedents and factors that may increase the risk for procrastination.

## 1. Procrastination and Stress: A Conceptual Review of Why Context Matters

Whether viewed as an occasional behaviour or a chronic behavioural tendency, procrastination is a common form of self-regulation failure that is linked to negative outcomes. Procrastination is often defined as the voluntary and unnecessary delay in the start or completion of important and intended tasks despite recognising there will be harmful consequences for oneself and others for doing so [1,2]. Yet, the harms from procrastination are not limited to those involving productivity. Research over the past two decades has documented that procrastination can also have wide-ranging and negative consequences for health and well-being [3], especially when it becomes a chronic behavioural pattern. For example, procrastination is associated with higher stress [4,5,6,7,8,9], use of less adaptive coping strategies [10], poor health behaviours [5,9,11,12,13], poor quality sleep [14,15], poor self-rated health [16], and a greater number of physical illnesses and symptoms [6,9,12,17].

Theoretical accounts position stress as having a central role for understanding the implications of procrastination for health and well-being. The procrastination–health model [5,12] posits that procrastination, whether momentary or as a behavioural tendency, negatively impacts health through the generation of unnecessary stress. This stress can be from stress-generating thoughts about unnecessary delay [4], and from the personal and social consequences of that delay [18]. However, a temporal mood-regulation perspective on why people procrastinate proposes that the experience of negative emotional states and difficulty in regulating them underly procrastination behaviour [2,19]. From these perspectives, it is clear stress is a negative emotional state that can be both a cause and a consequence of procrastination.

An important and often overlooked consideration for understanding the dynamic interplay of procrastination and stress is the role of context. If negative emotional states and their management are key for understanding when people procrastinate, then it is reasonable to expect that stressful contexts will increase vulnerability for procrastination, both for those who are prone to procrastinate and those who procrastinate infrequently. The aim of the current review is to summarise current theory and research to outline a new conceptual framework for understanding the role of context in procrastination, and specifically with respect to how stressful contexts can contribute to vulnerability for procrastination, using the COVID-19 pandemic as an example. The COVID-19 pandemic is a global health crisis involving the rapid and deadly outbreak of the coronavirus caused by the severe acute respiratory syndrome coronavirus 2 (SARS-CoV-2) virus starting in December 2019 [20], which caused widespread societal disruption globally.

Drawing on insights from coping and emotion regulation theory, I will also argue that this new stress context vulnerability framework is highly relevant for understanding procrastination during the COVID-19 pandemic. Moreover, it can be applied for understanding how and why risk for procrastination may increase in other stressful contexts, as well as provide insights into the approaches that might help mitigate risk.

## 2. Procrastination, Stress, and Emotion Regulation

Research over the past 25 years has provided evidence for the robust links between procrastination and stress. Much of this research has examined stress with the assumption it is a consequence of procrastination, whether occasional or more habitual. Procrastination is moderately and positively associated with perceived stress in samples of adolescents [21], university students [6,9,12,16,22], community-dwelling adults [5,8,23], and individuals with hypertension and cardiovascular disease [17]. This research is primarily cross-sectional, making it difficult to confirm the direction of influence. However, there are at least two studies which have documented that a tendency to procrastinate predicts higher perceived stress over time. In one longitudinal study of 379 undergraduate students, procrastination at Time 1 predicted higher stress one and two months later after accounting for the contributions of relevant Big Five personality factors and demographic variables [24]. A 9-month longitudinal study with over 3500 Swedish university students found similar results. Baseline procrastination predicted higher perceived stress at the follow-up after controlling for a set of key socio-demographic variables, as well as initial stress levels to address the issue of reverse causality [9].

Although research has not fully explored the reasons why procrastination contributes to the generation of stress, there is some evidence that intrapersonal processes which involve appraisal processes may account for this link. For example, ruminative thinking can amplify negative states and contribute to chronic stress by reactivating and sustaining an acute stressor [25]. When ruminative thinking is focused on procrastination, research has found that it increases both distress and procrastination [4,26]. Studies have also demonstrated that a tendency to procrastinate is associated with low mindfulness [16] and low self-compassion [22], two qualities that involve appraising challenging events in a less threatening manner and in turn reduce stress [27,28]. Not surprisingly, low mindfulness and self-compassion accounted for the link between procrastination and higher stress [10,22]. Lastly, a meta-analysis found that procrastination was associated with greater use of maladaptive coping strategies [10], and in one study, use of maladaptive coping accounted for the association between procrastination and higher stress [17].

Theory and evidence also suggest stress may be a precursor, and not just a consequence, of procrastination. A temporal mood regulation perspective posits that people procrastinate tasks which elicit negative emotional states as a means of regulating their immediate mood through task avoidance [2,19]. Negative states can arise from the nature of the task when it is inherently aversive or unpleasant (e.g., public speaking), or because the individuals’ interaction with the task generates difficult emotions including uncertainty, anxiety, or stress [29,30]. Indeed, research confirms that encountering tasks which are perceived as aversive or that generate negative emotions are a precursor to procrastination [26,31]. A temporal mood regulation view of procrastination takes this further. Difficult task-related emotional states are a necessary but not sufficient condition for procrastination—how one manages and regulates these emotions is also a crucial consideration. If short-term mood regulation is prioritised over long-term goals, then people will procrastinate as a means of making a positive hedonic shift by avoiding the difficult emotions associated with an aversive task [2]. In support of this perspective, randomised controlled trials and prospective research have demonstrated that interventions which improve emotion regulation skills are effective for reducing procrastination [32,33,34]. In short, procrastination can be understood as poor mood regulation in the form of avoidant coping that shifts the focus to “feeling good now” [35], but at the cost of reaching goals.

The relevance of a temporal mood regulation perspective for understanding the context dependency of procrastination also becomes clearer when we consider the approaches people use to manage negative emotional states. Stress is a particular negative emotional state in which the perceived demands and threat of a situation outweigh the resources available to manage these demands [36]. When people experience negative states such as stress this triggers efforts to make a hedonic shift from negative to less negative or more positive states [37]. People can employ a number of coping and emotion regulation strategies to accomplish this change. Adaptive strategies include changing the way they appraise the stressful situation, and approaching and taking action to change the situation, whereas less adaptive strategies focus on suppressing, avoiding, or denying the stressful situation and the emotions experienced [37,38]. From an emotion regulation perspective [37], whether the strategies are effective is often determined by whether the hedonic shift is maintained over time. Similarly, from a stress and coping perspective [36], adaptive coping strategies are those which reduce or remove the source of the stress. Alternatively, where this is not possible, adaptive coping strategies change the way the individual views or interacts with the stressor, such as through reappraisal. Successful coping efforts are therefore those which help reduce the experience of stress over time.

From both a practical and theoretical perspective, there are clear benefits to using adaptive rather than maladaptive emotion regulation and coping strategies to reduce stress. Such approaches reduce not only the stress in an effective and lasting manner, but also the negative psychophysiological impact it can have on immune functioning, nervous system reactivity, sleep, and well-being, if stress is not effectively managed [39,40,41]. Yet despite these benefits, people will sometimes use less adaptive strategies, such as avoidance, to manage stress.

There are several factors that determine whether individuals use adaptive or maladaptive coping strategies for managing stress. Coping choices are often based on situational factors, such as the specific demands of the situation, and the extent to which resources such as social support are available [36,38]. However, individual differences and personality traits can also predispose the use of certain coping strategies for better or worse [42,43].

As an avoidant behavioural tendency, chronic or trait procrastination is unsurprisingly associated with the use of maladaptive coping strategies, which can contribute to the dynamic and mutually reinforcing links with stress. For example, in one meta-analysis of 15 diverse samples including a total of 4357 participants, trait procrastination was positively associated with an index of maladaptive coping strategies (i.e., denial, self-blame, behavioural disengagement and substance use), and negatively associated with an index of adaptive coping strategies (i.e., active, planning, instrumental and emotional support seeking) [10]. A further analysis of four of the samples that included a measure of perceived stress found that maladaptive but not adaptive coping strategies accounted for the association between procrastination and higher stress. This evidence suggests that not only is chronic procrastination linked to the use of a range of maladaptive coping strategies to manage stress, but that the use of such strategies is another way in which procrastination can contribute to stress.

## 3. Stressful Contexts and Vulnerability for Procrastination

Despite this burgeoning evidence base and theory linking procrastination to higher levels of stress, as well as the reverse, the role of context in this potentially dynamic association has received relatively little attention. One notable exception is research examining how exposure to and availability of digital and technological distractions can exacerbate both stress and procrastination. For example, studies with large representative samples have demonstrated that unrestricted and poor control over Internet use amplifies the detrimental effects of procrastination on well-being, including stress [21,23]. Although the Internet is arguably not a stressful context per se, the pervasive availability of online activities and social media via smartphones, tablets, laptops, and personal computers in daily life does create a contextual backdrop of digital distractions that can provide an accessible means of procrastinating.

From the perspective of temporal mood regulation, contexts characterised by high levels of stress should be particularly relevant for understanding procrastination. If poor mood regulation in dealing with difficult task-related emotions explains *why* people procrastinate [2,19], then stressful circumstances and contexts can provide insights into *when* people may be more vulnerable to procrastinate. For example, relative to less stressful contexts, high-stress contexts present challenges that can contribute to ongoing and elevated levels of negative affective states, including feelings of stress and anxiety [44,45].

From this perspective, two key hypotheses can help explain why stressful contexts, such as the recent COVID-19 pandemic, can increase mood regulation and coping demands, and subsequently increase risk for procrastination. The ongoing stress experienced in such contexts reduces coping resources and capacities, as well as the threshold of tolerance for dealing with negative states. In short, both contextual influences promote procrastination because it is a low resource means of avoiding aversive and difficult task-related emotions. As will be detailed in the following sections, *coping depletion* and *phenotypic vulnerability* may explain the increased risk for procrastination for those prone to chronic procrastination, and for those who procrastinate infrequently. Figure 1 provides an overview of these vulnerabilities and the processes that may contribute to the dynamic linkages of procrastination and stress in high-stress contexts.

### 3.1. Coping Depletion Vulnerability

When individuals experience a single stressful event, the temporary nature of that event most often means its effects are likely to be acute, rather than lasting. After the experience of a single stressful event there is usually time for both physiological recovery (i.e., a return to physiological homeostasis) and psychosocial recovery (i.e., replenishing personal and social coping resources) from the disruptive effects of the stressor. This recovery period provides an opportunity to return to pre-event baseline levels of functioning. However, when a stressful event is ongoing, and/or there are repeated activations of the stressor [25], there is often little opportunity for immediate recovery. Ongoing use of coping resources to manage stress, including reappraising the stressor in less threatening ways or mobilising social support, may deplete these resources and/or the effort required to mobilise them.

Low-resource contexts can increase vulnerability to using coping strategies that are less adaptive for long-term management of stress and negative emotions, in part because such strategies have lower associated costs. Indeed, evidence indicates that coping strategies, such as distraction, provide immediate but temporary mood regulation, require fewer cognitive resources than strategies such as cognitive reappraisal, which is more cognitively demanding [46]. However, there is a trade-off of between resource costs and effectiveness. Strategies such as cognitive reappraisal provide more effective and longer lasting regulation of negative emotions [37].

The importance of considering low-resource contexts for how people regulate their emotions is also highlighted in the Strengths and Vulnerabilities Integration (SAVI) model [47]. Although the SAVI model addresses age-related changes in emotion regulation resources, its key tenets regarding the difficulties that individuals with fewer psychological and physiological coping resources experience when regulating high-arousal negative emotions, especially when sustained over time, are consistent with a coping resource depletion view of procrastination. Accordingly, against a backdrop of stressful and demanding circumstances, people may be more vulnerable to use procrastination as a means of emotion regulation for dealing with difficult task-related emotions due to depletion of their coping resources from ongoing stressors.

Coping depletion can also help explain why some individuals who might not normally procrastinate important tasks do so in the context of ongoing stressors. When faced the sustained demands of a stressful situation versus the occasional demands of a difficult or stressful task, limited coping resources may be directed towards the ongoing stressor rather than the difficult task. In this situation, procrastination may become an appealing choice for coping with task-related negative emotions. As an avoidant coping strategy, procrastination is less costly in terms of resource expenditure [46], and it provides immediate relief from the additional stress and negative emotions associated with the task [2,19]. However, as is the case with most avoidant coping strategies, the relief from procrastination is temporary, and the stress and negative emotions that were avoided not only return, but also can increase [9,26,48].

### 3.2. Phenotypic Expression Vulnerability

Another reason why stressful contexts might create vulnerability for procrastination is because they may lower the tolerance for stress and other negative affective states, especially among those who are prone to procrastination. This explanation is plausible if we consider chronic or habitual procrastination as a relatively stable personality trait reflecting difficulties in emotion regulation. Indeed, behaviour genetics studies with twins reared apart suggest that trait procrastination is moderately heritable (46%) at the genotypic level [49]. However, personality traits can also be expressed and observed to a greater or lesser degree depending on environmental factors that facilitate or hinder their expression [50]. Accordingly, phenotypic expression of the personality trait can vary, due in part to the influence of the situational characteristics and demands of the environment.

Stressful contexts and circumstances create a backdrop of increased negative states that require ongoing emotion regulation efforts to manage them. Such contexts can be particularly challenging for individuals prone to procrastination, who already struggle with emotion regulation. For example, in non-stressful circumstances and contexts, tasks that are associated with more intense negative emotions are more likely to be procrastinated [31], whereas when task-related negative emotions are less intense, there may be less risk for procrastination. However, in the context of ongoing background stressors, even less intense negative tasks may be perceived as more intense. This may be due in part to the cumulative effects of negative emotional experiences, and a perceived difficulty in being able to regulate emotions even in non-stressful circumstances. Indeed, research has demonstrated that negative emotional experiences can have a cumulative effect that can increase avoidance of negative states after each successive experience [51]. The threshold for tolerating negative emotions may therefore be lowered due to this cumulative effect. Accordingly, vulnerability for procrastinating tasks that might not normally be procrastinated increases, especially among individuals who are already prone to procrastination and are using task avoidance as a habitual means to regulate difficult emotions [19]. In this respect, stressful contexts create an environment that increases vulnerability for the phenotypic expression of trait procrastination.

High-stress contexts may also contribute to lowered tolerance for negative states because stress compromises behaviours that might otherwise bolster tolerance, and potentially creates a downward spiral of lower tolerance. For example, perceived stress is associated with poor sleep [52], with one population-based study finding that stress accounted for 24 percent of the variance in sleep quality [53]. Poor sleep quality in turn can increase reactivity to stress [54], and impair emotion regulation [55,56]. In one longitudinal study, sleep problems predicted lower tolerance for distress one year later, with becoming absorbed by negative emotions and experiencing stress as being unacceptable as two key facets of tolerance that were lowered [57]. Taken together, this research suggests that in stressful contexts that impair sleep, tolerance for negative states is reduced, thereby increasing vulnerability for using procrastination as means to cope with negative states.

Rather than being independent, it is likely that both coping depletion and phenotypic expression vulnerability interact in dynamic and mutually reinforcing ways. This proposition is reasonable if we consider that perceptions of whether or not someone can cope with the demands of a situation are influenced by subjective views of one’s own innate and/or current capacities to tolerate certain threats [42,58].

## 4. COVID-19 Stress and Vulnerability for Procrastination

The recent COVID-19 pandemic provides a real-world example of a stressful context that may have increased vulnerability to procrastination. This global public health crisis created disruption across a range of important life domains, leading to personal, societal, health-related, employment-related, and financial stressors [45]. Consistent with this proposition, a review of studies conducted during the early period of the pandemic found that procrastination increased in a number of different contexts, notably in academic settings [59]. For example, one study conducted during the COVID-19 pandemic with a sample of almost 9000 university students found that higher perceived stress increased students’ procrastination [60]. Similarly, a scoping review of procrastination during the pandemic found that procrastination was associated with higher levels of distress [61]. These findings are not surprising given the effects of the pandemic were pervasive and contributed to stressful changes that affected people’s daily lives in multiple ways.

As shown in Figure 2, there were four key sources of stress during the COVID-19 pandemic. In addition to the looming health threats associated with the pandemic, and adjusting to new regulations for how to stay safe in public, many individuals experienced stress due to changes in their working routines, a decrease in their level of social interactions, changes in the delivery of academic work, and even threats to their financial security as a result of the pandemic [59,62,63]. For some individuals, the stress due to fears about becoming infected and from the socio-economic impacts of the pandemic became so extreme that some clinicians suggested that it reflected a new COVID stress syndrome [64]. Taken together with the looming uncertainty of not knowing when (or if) the pandemic and the associated health threats and changes to lifestyle, social relationships, financial security, or working conditions would end, it is clear that the challenges of the pandemic created ongoing background stress for many individuals. This stress created increased coping demands and potentially a situation of *coping overload* that subsequently contributes to coping depletion. Additionally, the cumulative negative emotional experiences from the impacts of COVID-19 across various life domains, and the deleterious effects of stress on sleep quality, are two plausible ways in which COVID stress lowered the tolerance for negative states (See Figure 2).

The following sections outline the key primary (health threats) and secondary (social isolation, remote working, and financial insecurity) sources of stress during the pandemic and how they potentially contributed to procrastination via coping depletion and/or phenotypic expression vulnerability.

### 4.1. Health Threats

Perhaps the most ubiquitous and direct way that the COVID-19 pandemic created an ongoing context of stress was through its very real and looming threats to the health of individuals and their family members. At the time of this writing, over 758,390,546 people had been infected with COVID-19, and 6.8 million people had died due to coronavirus [45]. The swift spread of the coronavirus, associated high mortality rates, and lack of an effective treatment, created a global public health crisis that taxed the coping resources of individuals, health-care systems, and society. Not surprisingly, concerns about the threat of infection to self and others was a primary source of ongoing stress during the pandemic [45,65]. For those working in health-care settings or service industries that involved frequent public contact, the threat of infection was especially salient and contributed to heightened levels of stress [66,67]. In addition, media communications were dominated by frequent reports of rising infection and death rates, which likely amplified stress through repeated exposure to the public health crisis posed by the COVID-19 pandemic [68]. Similarly, meta-analyses of studies from around the world found that levels of fear of COVID were high across the world during the pandemic [69], and that fear of COVID was robustly linked to higher stress [44].

Although there has been little research directly examining how health threats may have impacted procrastination during the pandemic, findings from at least one study support the proposition that health-related stress may have increased vulnerability for procrastination. In a large sample of undergraduate students surveyed during the COVID-19 pandemic, high levels of fears of COVID-19 were linked to more frequent self-reported procrastination [70]. As a high-arousal negative emotional state, fear, like stress, requires coping strategies and resources to manage it. Because general rather than domain-specific procrastination was measured in this study, these findings suggests that managing COVID-related stress and fear depleted the coping resources that would be otherwise used to manage the negative emotions associated with difficult or aversive tasks, and therefore increased vulnerability for procrastination.

### 4.2. Social Isolation and Loneliness

One important and prevalent form of secondary stress during the COID-19 pandemic was social isolation. The social isolation and restriction of social activities resulting from public health and government recommendations and implementation of social distancing measures, lockdowns, and stay-at-home measures to limit social contact and the spread of the coronavirus, was a prominent source of pandemic-related stress [71]. Those who depended on social resources [72], and who experienced a significant increase in social isolation due to the pandemic, including university students [73,74], were particularly vulnerable to experiencing stress related to social isolation. Loneliness, a distinct but related concept to social isolation that involves the subjective feelings of lacking the quality of relationships that one desires [75], was also heightened for many during the pandemic [76,77]. Similar to social isolation, loneliness is well known to predict stress over time as well as increase reactivity to stress [78,79]. Social interactions are widely recognised as an important and effective coping resource that can mitigate stress [39], especially in stressful times such as during the COVID-19 pandemic [80]. Given this, a decrease in meaningful social connections and contact can be stressful because it is tantamount to having fewer coping resources to manage ongoing stressors.

There is some evidence that loneliness is linked to procrastination [8,81], although the direction of the relationship is not clear. For example, one longitudinal study found that a tendency to procrastinate is linked to greater loneliness over time [9]. However, in another study, an 8-week psychoeducation program that included components aimed at reducing loneliness via increasing social connection was effective for reducing academic procrastination [82]. It is likely that the links between social isolation, stress, and procrastination are complex and mutually reinforcing. Nonetheless, the proposition that the backdrop of stress from social isolation during the pandemic contributed to procrastination is supported by the findings from at least one study. In a survey of employees during the pandemic, being socially isolated from colleagues in the workplace was a source of stress that negatively impacted perceived workplace productivity [83]. If we view low productivity as a potential proxy for procrastination, this evidence is consistent with a coping depletion vulnerability explanation for procrastination during the pandemic.

### 4.3. Remote Working

Ostensibly, it might be expected that procrastination would increase under remote working conditions because of the increased availability of distractions. Indeed, many working environments are structured and include dedicated working space that is set up to promote focus and reduce unnecessary distractions. From a mood regulation perspective of procrastination [2,19], distractions provide opportunities for procrastination but are not its core cause. Instead, the lack of structure in remote working situations and uncertainty from working under new conditions are conditions that can generate stress, and in turn trigger procrastination as a means of coping with that stress, e.g., [31,84]. This may be especially true if coping resources are depleted due to dealing with other stressors during the pandemic.

Research examining the links between COVID-19 remote-working stress and procrastination is scant, but aligns with a coping depletion explanation of procrastination during the pandemic. In one qualitative study, employees who had to work remotely noted that procrastination was a key challenge when working from home [85]. However, in a follow-up survey, employees working remotely who had higher levels of perceived social support also procrastinated less. Given that social support is a powerful and well-established resource for reducing stress [39], this finding is consistent with the proposition that increased stress whilst working remotely may have contributed to procrastination.

### 4.4. Financial Insecurity

In addition to the challenges from remote working, financial insecurity was another work-related secondary source of stress during the pandemic that affected many people. Social distancing and lockdown conditions negatively impacted a number of business sectors, including retail, hospitality, travel and tourism, and beauty and grooming services, with small businesses being particularly affected [86,87]. Suspension of services and business closures resulted in people being laid off work or losing their jobs, and contributed to financial insecurity that increased distress, especially for working parents [88]. Not surprisingly, longitudinal research comparing pre-pandemic to during-pandemic perceived stress found that economic stressors were a key contributor to the increases in stress reported during the pandemic [62].

The financial stress that many individuals faced during the pandemic could have increased vulnerability for procrastination in several ways. For those who already were prone to procrastination, the financial stressors of the pandemic may have increased their financial procrastination. Several studies have noted that chronic procrastination is linked to unnecessarily delaying important financial behaviours. For example, people who frequently procrastinate tend to postpone saving for retirement, are less likely to save, engage in last-minute shopping, and miss bill deadlines [89,90]. Financial activities, such as retirement planning and saving, are heavily influenced by a range of negative emotions, including dread, worry, and anxiety, which may be difficult for some individuals to manage [91,92]. The financial stress of the pandemic may therefore have made it even more difficult for those prone to procrastination to engage with financial behaviours, thus increasing their financial procrastination. Additionally, the widespread implications that financial stressors have for long-term security and daily living create a pervasive backdrop of stress that may have depleted the psychological resources needed to cope with the negative emotions linked to challenging tasks that were unrelated to finances. In this respect, financial stress during the pandemic likely contributed to both phenotypic expression and coping depletion vulnerability for procrastination.

## 5. Applications of a Stress Context Vulnerability Model of Procrastination

The preceding sections outlined the relevance of a stress context vulnerability view of procrastination during the COVID-19 pandemic. Importantly, this framework can be applied to understand vulnerability for procrastination in other contexts involving ongoing or prolonged stressful circumstances that might deplete coping resources and/or lower tolerance for managing negative task-related emotions. One such stressful context that may increase the risk for procrastination due to depletion of coping resources is that of living with a chronic illness or health condition.

Chronic illnesses can present a number of challenges, such as managing pain, fatigue, unpredictable symptoms, and functional limitations, that can tax coping resources and create vulnerability for procrastination of general and health-specific tasks. Research provides some support for this proposition. Research indicates that people with chronic pain report higher levels of stress compared to healthy controls [93,94], and that they tend to use more avoidant coping strategies for managing stress [95]. Similarly, a study of people with inflammatory bowel disease found that as disease severity, a key determinant of stress, increased, so did use of avoidant coping strategies [96]. Given the theoretical links between procrastination and avoidant coping, this evidence provides some support for the notion that the stressful context of living with a chronic health condition may increase vulnerability for procrastination.

Psychosocial stressors, such as those that were prominent during the COVID-19 pandemic, can also create a backdrop of heightened stress that may increase coping depletion and/or phenotypic vulnerability for procrastination for some individuals. For example, employment and financial insecurity are well-known sources of chronic stress [97,98]. Similarly, social isolation and loneliness are societal challenges that can create ongoing background stress [78,79,99] that may weaken emotion regulation capacities as well as reduce perceived social coping resources.

In addition to the contexts outlined above, any circumstances that create prolonged periods of stress, and especially those which do not have a clear end point, are potential contexts that can increase vulnerability for procrastination. For example, being a caregiver for an older adult family member who is ill is well-known to be linked to chronic stress [100]. Other stressors to consider that can create a backdrop of ongoing stress include immigration, which can create acculturation stress [101], especially for international students studying abroad [102], and workplace bullying, which has been identified as a serious and often severe form of workplace stress [103].

## 6. Implications and Future Directions

The stress context vulnerability model provides a novel perspective on when and who may be more vulnerable to procrastination, as well as the dynamic and reciprocal processes involved. By proposing that reductions in coping resources and/or tolerance for negative affective states are key reasons why stressful contexts can increase vulnerability for procrastination, this new model also highlights approaches for addressing procrastination in stressful circumstances. For example, bolstering coping and emotion regulation resources and skills to better manage ongoing stressors may be effective approaches for short-circuiting the stress–procrastination cycle.

Approaches that are known to both manage stress and reduce procrastination could be particularly beneficial for addressing vulnerability for procrastination during times of high stress. For example, self-compassion, taking a kind, accepting, connected, and mindful approach to personal flaws and setbacks [104], is an effective coping resource that is known to reduce stress [105,106], and is linked to lower procrastination [22]. Importantly, self-compassion can be cultivated through short programmes and/or self-help exercises [107], making it an attractive approach for bolstering resilience in low-resource contexts.

Mindfulness is another approach for managing negative emotions that has demonstrated effectiveness for reducing both stress and procrastination. Evidence supports the effectiveness of mindfulness and mindfulness-based practices for reducing stress in general contexts [108], and for those working in stressful contexts [109]. Not surprisingly, research has found that a tendency to procrastinate is associated with low mindfulness [16], and that higher mindfulness is linked to less procrastination over time [110]. More importantly, interventions that increase mindfulness have been found to also reduce procrastination, in part by reducing negative emotions [34]. Providing mindfulness training and resources to those who are experiencing stressful circumstances and at risk for procrastination may have dual benefits. Research examining the effects of mindfulness on procrastination during stressful circumstances would be well-placed to test this proposition.

Aside from mindfulness and self-compassion, future research could test other approaches such as the provision of emotion regulation or coping skills training. Interventions that focus on improving emotion regulation skills, which often involve the above approaches, are likely to help as well. For example, one randomised control trial found that a 2-week systematic emotion regulation skills training focused on tolerating and modifying negative emotions significantly reduced procrastination at the follow-up [33]. There is also some evidence that interventions for improving coping resources, such as increasing social support, and modifying coping styles to be more-approach-oriented and less avoidant, can be effective [111]. Such approaches could be particularly effective for reducing procrastination in high-stress contexts, as they have potential for directly addressing vulnerability due to coping depletion and due to the poor tolerance and regulation of negative emotions that characterise phenotypic expression vulnerability.

## 7. Conclusions

By considering the role of context in vulnerability for procrastination, this new stress context vulnerability framework underscores the need for taking a more compassionate view of the antecedents and factors that may increase the risk for procrastination. Whereas previous views of the reasons for procrastination have focused mainly on pre-existing individual differences in self-regulatory capacities, and specifically emotion regulation weaknesses, this new framework suggests that culpability for procrastination does not solely or always rest within the individual. Instead, this view of procrastination suggests that there also needs to be consideration of any ongoing stressful circumstances that might make mood regulation more challenging, and therefore increase vulnerability for procrastination. Taking such stressful contexts into account can help identify when and who might be most vulnerable to procrastinating, as well as provide insights into what support might be needed to address and prevent procrastination.

## Figures and Tables

**Figure 1 ijerph-20-05031-f001:**
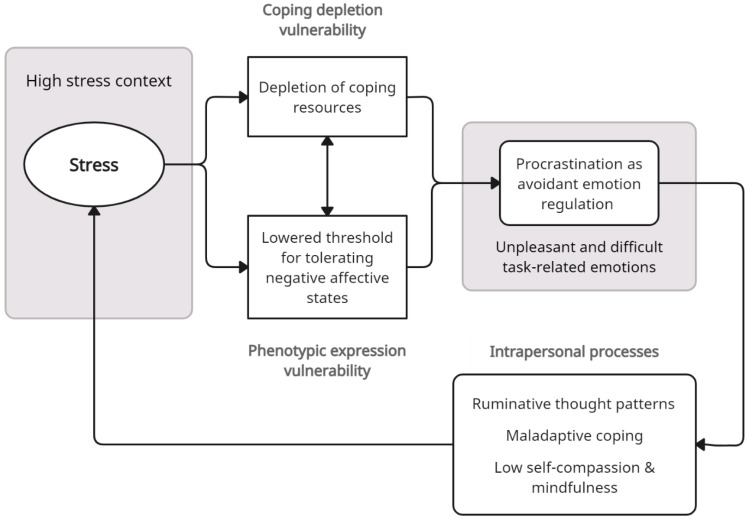
Stress context vulnerability model of procrastination. Note: High-stress contexts are proposed to increase vulnerability for procrastination via depletion of coping resources and lowering tolerance for negative states. Procrastination is used as an avoidant coping strategy to manage negative task-related emotions, and subsequently generates further stress through intrapersonal appraisal processes, thereby amplifying contextual stress in a cyclic and dynamic manner.

**Figure 2 ijerph-20-05031-f002:**
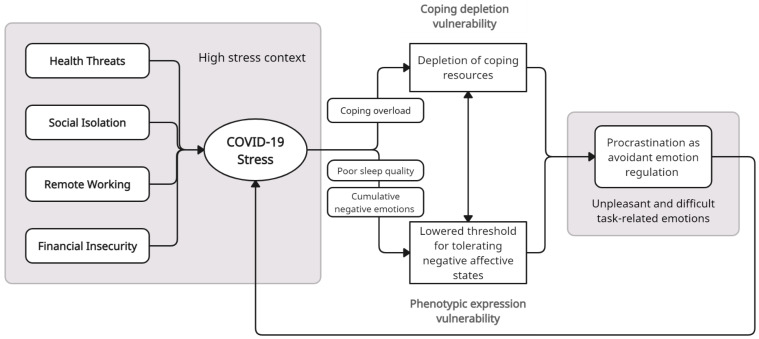
Stress context vulnerability model of procrastination applied to the COVID-19 pandemic. Note: Four main sources of stress during the COVID-19 pandemic are posited to contribute to vulnerability for procrastination through coping overload, which depletes coping resources, and by compromising sleep quality and a cumulation of negative states, which lower the tolerance for negative states. The procrastination that results further amplifies this stress in a cyclic manner.

## Data Availability

Not applicable.

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
