# Peer review of "Procrastination and Stress: A Conceptual Review of Why Context Matters"

_ijerph, 2023, doi:10.3390/ijerph20065031_

Round 1

Reviewer 1 Report

The purpose of this manuscript is to discuss the relationship between procrastination and stress from different perspectives to build a new conceptual framework, using COVID-19 as an example of a stressor.

The impression: The topic may be interesting, but the manuscript is very hard to read. The structure of the sentences is generally complex. The author needs to revise to improve the readability of the manuscript.

In addition, there is a need to revise punctuation to improve the structure of the sentences.

Although it is very appreciative that the author is referencing a statement, sometimes with multiple references within a sentence, people may lose focus.

I noticed that the reference style does not match the journal requirements.  

Other comments:

The author needs to revise extensively syntax and punctuation. For instance, line 191-192: there is a use of "that" 3 times, which weakens the structure of the sentence. Instead of saying "a singular event that is stressful" this could be reworded as "a stressful singular event".

 Since the purpose of the review is to address procrastination using COVID-19 as a model, it would be nice to give some more detailed information in figure 2, which does not differ much from figure 1 except for listing the potential stressors of COVID. Give more specifics about the lowered threshold for tolerating a negative state ( for instance sleep, diet, exercise changes) and other information to add value to the work. These factors should be different during the pandemic. Figure 2 needs more details since this is the scope of the manuscript.

Typos and grammar mistakes: line 106 "has" should be "have"; line 268 real-world, not word; line 263 repeated words. line 440: "those" should be followed by "who" not by "that". 

The stress-vulnerability model addresses environmental stressors and their management including alcohol and drug abuse. The manuscript does not address these major points, especially their impact on stress during COVID which connects to my previous point. 

Reviewer 3 Report

The manuscript provides review on procrastination and stress. This article is important from a theoretical point of view for professionals who work with the consequences of stress and procrastination. However, some points are need to be improved. I believe that once revisions are made to the article, it would constitute an important contribution to the literature. I give some comments below.  

     1.      Figures notes are missing.

2.     Literature includes less than half of recent links (within the last 5 years). You should add recent links.

3.     It would be useful to discuss recent reviews on the problem of stress and procrastination in the context of the COVID-19 pandemic, for example https://www.mdpi.com/2076-328X/12/2/38, https://knepublishing.com/index.php/KnE-Social/article/view/8220, https://ijrpr.com/uploads/V3ISSUE2/ijrpr2603-a-critical-review-on-parental-involvement-academic-procrastination.pdf, http://seminar.uad.ac.id/index.php/ICMPP/article/view/6930.

Round 2

Reviewer 1 Report

The author has done a great job editing the manuscript, it reads much better and the presentation of the information is more interesting now. I have no further comments